# The effects of mycorrhizal colonization on phytophagous insects and their natural enemies in soybean fields

**Élisée Emmanuel Dabré**[1]*, **Soon-Jae Lee**[2], **Mohamed Hijri**[1,3], **Colin Favret**[1]

**1** Department of Biological Sciences, Biodiversity Centre, Plant Biology Research Institute, University of Montreal, Montréal, Québec, Canada, **2** Department of Ecology and Evolution, University of Lausanne, Lausanne, Switzerland, **3** AgroBioSciences, Mohammed VI Polytechnic University (UM6P), Ben Guerir, Morocco

* dabremanuel@yahoo.fr

**Data Availability Statement:** All relevant data are within the paper and its Supporting Information files.

**Funding:** This work was supported by the Natural Sciences and Engineering Research Council of

## Abstract

The use of belowground microorganisms in agriculture, with the aim to stimulate plant growth and improve crop yields, has recently gained interest. However, few studies have examined the effects of microorganism inoculation on higher trophic levels in natural conditions. We examined how the diversity of phytophagous insects and their natural enemies responded to the field-inoculation of soybean with a model arbuscular mycorrhizal fungus (AMF), *Rhizophagus irregularis*, combined with a nitrogen-fixing bacterium, *Bradyrhizobium japonicum*, and a plant growth-promoting bacterium, *Bacillus pumilus*. We also investigate if the absence or presence of potassium fertilizer can affect this interaction. We found an increase in the abundance of piercing-sucking insects with the triple inoculant irrespective of potassium treatment, whereas there were no differences among treatments for other insect groups. A decrease in the abundance of the soybean aphid, *Aphis glycines*, with the double inoculant *Rhizophagus* + *Bradyrhizobium* was observed in potassium enriched plots and in the abundance of *Empoasca* spp. with potassium treatment independent of inoculation type. Although it was not possible to discriminate the mycorrhization realized by inoculum from that of the indigenous AMF in the field, we confirmed global negative effects of overall mycorrhizal colonization on the abundance of phytophagous piercing-sucking insects, phytophagous chewing insects, and the alpha diversity of phytophagous insects. In perspective, the use of AMF/Rhizobacteria inoculants in the field should focus on the identity and performance of strains to better understand their impact on insects.

## 1. Introduction

Plant beneficial soil microbes, including mycorrhizal fungi and plant growth-promoting rhizobacteria (PGPR), have long been studied and applied for their positive effects on plant growth, nutrient mobilization, and agricultural product yield [1,2]. Arbuscular mycorrhizal fungi (AMF), of the phylum *Glomeromycota* [3,4], form one of the most widespread symbiotic associations with plant roots and constitute an important functional group in terrestrial ecosystems

Canada (NSERC) Discovery grants to CF (RGPIN 2017-06287) and to MH (RGPIN-2018-04178). EED received a scholarship from the Islamic Development Bank to support education fees and living allowances. Premier Tech Biotechnologies provided the seeds and inoculants and, in collaboration with La Coop Fédérée, the access to fields trials in for data collection.

**Competing interests:** The authors have declared that no competing interests exist.

[5]. They are obligate biotrophs that form symbiotic associations with more than 80% of vascular plant species [6,7]. Whereas the autotrophic plant delivers photoassimilates to the fungus, the fungal partner, in return, improves water and nutrient uptake, especially of phosphorus [8].

Plant growth-promoting rhizobacteria are also one of the major microbial groups that interact with plants. There are those that are obligatory symbiotic with the plants, for example rhizobia and legumes, and those that are free-living near, on, or even within the plant organs, these latter known as endophytes [9]. Rhizobia are a polyphyletic group of Gram-negative bacteria that are associated with most legumes by forming root nodules and that facilitate plant growth by fixing nitrogen [10]. Free-living PGPR can promote plant growth directly or indirectly [11]. Direct effects of these bacterial associates are related to production of plant growth regulators (phytohormones, antioxidants and enzymes) or improvements in nutrient uptake [12]. Indirect effects are related to the production of metabolites, such as antibiotics or siderophores that decrease the growth of phytopathogens and other deleterious organisms [13,14].

Soybean, *Glycine max* (L.) Merr., is the most important agricultural legume for oil and protein production [15]. Legumes can form tripartite symbiotic associations with nodule-inducing rhizobia and AMF, that may benefit growth, development, and the uptake of both phosphorous and nitrogen [16–19]. It was shown in soybean and common bean (*Phaseolus vulgaris* L.) that the co-inoculation of AMF and rhizobium improved production, efficiency of photosynthesis, nodulation and, especially, an increase in both phosphorus and nitrogen concentrations [20,21]. In addition, some studies showed that co-inoculation of soybean with PGPR of *Bacillus* species and the rhizobia *Bradyrhizobium japonicum*, increases nodulation and nitrogen fixation [9,22].

Chemical inputs in the form of nitrogen, phosphorus and potassium fertilizers are used to increase agricultural productivity. Their use generally increases potentially pestiferous herbivore populations [23], although in some cases they can reduce the density of these insects [24]. For example, the use of potassium fertilizer in soybean reduces aphid populations by limiting the availability of amino acids [25]. In sustainable, low-synthetic-input agricultural cropping systems, the use of microorganisms as inoculants, in combination with certain fertilizers, may help maintain soil fertility and plant health [17]. As a consequence of the symbiosis, these microorganisms can change host plant feature for insect herbivores through their impact on plant nutritional quality and/or by priming effects that lead to enhanced inducible and constitutive plant defences [26,27]. Previous studies have shown that AMF-induced changes in plant traits positively or negatively affect the individual performance of aboveground phytophagous insects [26,28,29]. At the community level, AMF positively affected piercing-sucking insects and specialist chewing insects, but generalist chewers are negatively affected [26,30]. As with AMF, rhizobacteria can also influence plant–herbivore interactions, but the consequences depend on the identity of the plant and insect species and the degree of insect specialism [31,32]. For example, in a garden experiment, a study on rhizobacteria-insect communities showed that rhizobia affected the abundance of chewing insects, while no effect was observed with the sap feeders (piercing-sucking insects) [33].

Little is known about the effects of AMF and PGPR (and/or Rhizobia) on foliage-feeding insects in natural agricultural conditions [32]. Most studies to date have been undertaken under controlled conditions in the laboratory or green-house [27,28,33], and by using methods to eliminate indigenous microorganisms from experimental fields [34,35]. Also, microorganism-plant-insect interactions impact not just the herbivores, but also higher trophic levels such as their natural enemies [36,37].

We examined the effects of a combined inoculation of the AMF *Rhizophagus irregularis* (synonyms: *Rhizoglomus irregulare*, *Glomus irregulare*, *G. intraradices*) isolate DAOM 197198

(PTB 297), the Rhizobia *Bradyrhizobium japonicum* strain PTB 162, the PGPR *Bacillus pumilus* strain PTB180 and potassium fertilizer on the phytophagous insects of soybean and their natural enemies under field cropping conditions.

As the ecological consequences of inoculants in the field are poorly understood and may not be easily predicted [38], in this study we sought to anticipate possible undesired effects of the inoculation of AMF and rhizobacteria on the communities of phytophagous insects. The co-inoculation of the three microorganisms can affect the host plant growth, development, and nutritional status directly and indirectly by the interaction with local microbial community. Therefore, we hypothesized that: 1) the inoculation of AMF and rhizobacteria will increase the abundance and species richness of functional groups such as piercing-sucking and chewing insects of host plant; and that 2) the abundance and richness of these insects will be correlated with the degree of mycorrhizal root colonization. To test these two hypotheses, we conducted an experiment on two agricultural fields of soybean.

## 2. Materials and methods

### 2.1. Living material

Seeds of the soybean cultivar AURIGA and microorganism inoculants were supplied by Premier Tech (Rivière-du-Loup, Quebec, Canada) as follows:

**AGTIV®SOYBEAN** powder based on a mixture of *R. irregularis* isolate DAOM197198 (PTB 297) at a dose of 2750 viable spores per gram of product and *B. japonicum* PTB 162 at a dose of $2.5 \times 10^9$ cells per gram of product. Another inoculant consisted of mixture of **AGTIV®SOYBEAN** and *Bacillus pumilus* strain PTB180.

### 2.2. Experimental design

The study was conducted from May to September 2017, in two fields located at Varennes (45.693˚ N, 73.365˚ W) and Saint-Simon (45.681˚ N, 72.856˚ W), Quebec, Canada. The climate type at both sites, approximately 35 km from each other, is temperate-cold. The growing season typically lasts 5 months, from May to September, with July being the hottest month and August being the wettest. At Varennes, the average temperature is 20.1˚C with maximum of 24.8˚C and minimum of 15.4˚C and the average precipitation is 69.9 mm. At Saint-Simon, the warmest month recorded an average temperature of 19.8˚C with maximum of 24.8˚C and minimum of 14.7˚C and an average rainfall of 74.18 mm (See https://climat.meteo.gc.ca/ historical_data/search_historic_data_f.html). Sowing took place at Varennes and Saint-Simon on May 20 and 25, respectively. Seeds were precoated with substrate containing inoculants in the seed drill before sowing with an application dose of 300g per hectare. The experiment at each site consisted of a factorial design of three inoculant treatments (control [C], double inoculation with *R. irregularis* and *B. japonicum* [MR] and triple inoculation with the addition of *B. pumilus* to the double inoculant, [MRB]) combined with two treatments either with or without potassium fertilizer ([K-] and [K+]). So, we had 3 treatments per each potassium level/ block replicated 8 times, for a total of 48 plots per site. This potassium fertilizer (NPK [0–0– 60]) of 80 units was applied at a dose of 240 g per plot at Varennes and 175 g per plot at Saint-Simon following soil analysis one day before sowing.

At Varennes, each plot measured 6 m x 3 m, contained 4 rows of seedlings with 75 cm between adjacent rows, and an 8 m spacer separated each block. The previous crop had been wheat in this site and soybeans were grown alongside the trial during the growing season. On June 25, 2017, two herbicide treatments, Reflex® (Fomesafen) at a dose of 1L/ha and Pursuit® (Imazethapyr) at a dose of 0.312 L/ha were applied before the insect sampling. Also, a foliar Crop Booster (15-3-6 foliar spray fertilizer) was applied at 2 L /ha.

At Saint-Simon, each plot measured 5 m x 1.44 m, with row spacing of 36 cm, and a 2 m space separated each block. The previous crop had been maize, and soybeans were also grown alongside the trial. At this site, three herbicides were applied just before sowing: Dual II Magnum (S-metolachlor & R-enantiomer) at 1.75 L/ha, Pursuit® (Imazethapyr) at a dose of 0.312 L/ha and FirstRate (cloransulam-methyl) at a dose of 20,8 G/ha. Despite the use of herbicide, we noticed the growth of weeds at the early flowering stage (R1) on most of the 4 rows going towards a water canal located at 15 m.

The two sites were test sites of Premier Tech, which provided us the seeds and inoculants and they belong to Sollio Cooperative Group (formerly called La Coop fédérée), a cooperative of agricultural producers in Québec.

## 2.3. Insect trapping and sampling

We sampled insects at Varennes and Saint-Simon on July 4 and 11, respectively (active growth/early flowering stage). Three sampling methods were: yellow pan traps, pitfall traps, and D-Vac aspiration [39]: when the leaf-blower motor is activated, it rotates a fan that creates a flow of air through the tube to draw insects into a collecting bag attached to the end of a 1 m PVC tube.

In each plot, two pan traps and two pitfalls [40] were placed 3 to 5 m apart on the ground within the two center rows so that each trap was approximately 1.5 m from the edge of the plot and 2.25 m from another pan or pitfall in the adjacent plot. Each trap was filled approximately ¾ with water and a few drops of unscented dishwashing detergent to reduce the surface tension [41]. Traps were installed at 9:00 am the day before and recovered 24 hours later. The D-Vac sampling was carried out on the central rows of each plot for 1 minute before the traps were collected. Following each sampling, insects were either kept in the freezer [42], or stored in polyethylene bags (Whirl-Pak®) in 75% ethanol until identification [41].

## 2.4. Arbuscular mycorrhizal colonization measurement

Two individual plants with their roots and rhizosphere soils from the two central rows in each plot were uprooted randomly six weeks after sowing [43]. Roots were kept in polyethylene bags (Whirl-Pak®) with 50% ethanol and their cleaning was done according to the method described by Antunes et al. 2006 [44]. To estimate the mycorrhizal colonization rate, all the root samples were heated in 10% (wt/vol) KOH solution at 70˚ C for 1 hour before staining with a 5% solution of black ink (Sheaffer®) in 5% acetic acid solution for 20 min at 70˚ C. Roots were then cleaned for 40 min with acidified water (a few drops of 5% acetic acid in water) to remove excess of ink. These roots were mounted on slides and observed under an optical microscope at 100x magnification. The percentage of the root colonized was measured on structures like hyphae, vesicles and AMF arbuscules with the grid-line intersect method [45,46]. We assume that at plot level, each plant is more or less equally influenced by native fungi in its close environment. In this case, the measured mycorrhizal colonization rate is representative of the plot.

## 2.5. Insect identification

We sorted the insect specimens to order and family, when possible, and then into operational taxonomic units (OTUs) or morphospecies [47]. For this purpose, each OTU was photographed and several morphological characters were used for their characterization [47,48]. For Microhymenoptera, we favored the wing pattern, shape and length of antennae, whereas in Diptera we relied more on the wing pattern. For aphids (Hemiptera: Aphididae), we focused on the size and shape of the cornicles, the cauda, and sclerotization of the body. We identified

**Table 1. Number of specimens and operational taxonomic units (OTUs) of insect functional groups depending on the type of trap, D-Vac, pan trap (PT), and pitfall (PF), used during sampling in soybean fields.**

| Functional groups | Order | No. OTUs | Varennes | | | Saint-Simon | | | Total |
|---|---|---|---|---|---|---|---|---|---|
| | | | D-Vac | PT | PF | D-Vac | PT | PF | |
| *Aphis glycines* | Hemiptera | 1 | 57 | 0 | 17 | 213 | 0 | 1 | **288** |
| Other aphids | Hemiptera | 12 | 14 | 263 | 0 | 27 | 125 | 8 | **437** |
| **Aphids subtotal** | | **13** | **71** | **263** | **17** | **240** | **125** | **9** | 725 |
| *Empoasca* spp. | Hemiptera | 1 | 450 | 19 | 5 | 124 | 1 | 3 | **602** |
| Other piercing-sucking insects | Hemiptera, Thysanoptera, Coleoptera | 6 | 10 | 236 | 14 | 7 | 100 | 0 | **367** |
| **Piercing-sucking insects subtotal** | | **20** | **531** | **518** | **36** | **371** | **226** | **12** | **1694** |
| Chewing insects | Coleoptera | 16 | 40 | 181 | 22 | 7 | 35 | 16 | **301** |
| **Phytophagous insects subtotal** | | **36** | **571** | **699** | **58** | **378** | **261** | **28** | **1995** |
| Natural enemies of aphids | Coleoptera, Diptera, Hemiptera, Hymenoptera | 26 | 100 | 127 | 500 | 74 | 45 | 512 | **1358** |
| Natural enemies of other insects | Hymenoptera | 47 | 15 | 210 | 0 | 11 | 245 | 0 | **481** |
| **Natural enemies subtotal** | | **73** | **115** | **337** | **500** | **85** | **290** | **512** | **1839** |
| Other insects | Coleoptera, Diptera, Hemiptera, Hymenoptera, Orthoptera | 41 | 357 | 787 | 302 | 515 | 596 | 251 | **2808** |
| **Total** | | **150** | **1043** | **1823** | **860** | **978** | **1147** | **791** | **6642** |

the soybean aphid, *Aphis glycines* Matsumura (Hemiptera: Aphididae) [49], and *Empoasca* spp. (Hemiptera: Cicadellidae) [50], although this latter was probably mostly *Empoasca fabae* (Harris). Voucher material is deposited at the University of Montreal's Ouellet-Robert Entomological Collection.

To evaluate insect abundance and species richness, we focused on insects that may have a direct and indirect impact on the crop, namely phytophagous insects and their natural enemies. The insects were then classified into functional groups based on their feeding mode (for phytophagous insects, piercing-sucking and chewing insects) and mode of life (for natural enemies) [51] (Table 1).

## 2.6. Statistical analysis

All analyses were performed with R (version 3.5.3) (R Core Team 2019). The Shannon index (-diversity), that relates the number and relative abundance of species in each elementary plot, was calculated using the formula:

$$H' = -\sum_{i=1}^{s} \frac{n_i}{N} ln \frac{n_i}{N}$$

Where H' = Shannon's index of diversity; s = total number of species; $n_i$ = number of individuals of species i; N = total number of individuals of all species; ln = the natural logarithm [52]. To examine the effects of inoculants and potassium on crop yield, mycorrhizal colonization rate of roots, abundance (total number of specimens per OTU), and alpha diversity (Shannon index) per treatment for each functional group of insects, we used a linear mixed models (LMMs) with the *lmer* function in the package *lmerTest* [53].

Inoculant treatments (C, MR, MRB) and potassium ([K-], [K+]) were considered as fixed effects, while the block as the random effect, and they were crossed in the model. Other parameters (yield, abundance, Shannon index, mycorrhizal colonization rate) were the response variables used individually in the LMMs. The normality of the distribution and the homogeneity of the variance were tested by the Shapiro-Wilk and Levene's tests, respectively. When non-normal distribution and heteroscedasticity were observed, we used the function "sqrt" (square root) or "log" to transform the variable response before modeling [54]. When the model is

established, we applied the function "Anova" in the package "car" to test the difference among treatments. When significant differences were observed, *post hoc* test was applied with Tukey's honest significant difference (HSD) in package "mutlcomp".

To assess the influence of AMF colonization, we evaluated the correlation between various insect abundances or alpha diversity parameters and the mycorrhizal colonization measured in different plots. As the data did not meet the assumptions of normal distribution, we applied Kendall correlation analysis.

## 3. Results

### 3.1. Effects of AMF and rhizobacteria inoculation on crop yield and mycorrhizal colonization

Inoculation and fertilizer had no effects on root mycorrhizal colonization and grain yield at Varennes and Saint-Simon (S1 Table). There were no significant differences between inoculant treatments (C, MR, MRB) for AMF colonization of roots or crop yield irrespective of potassium (Table 2). We did not directly measure the effect of the inoculation on the bacterial community or on the taxonomic diversity of the microbial community in general.

### 3.2. Observed insect abundance and species richness among different inoculation treatments

A total of 6642 insects were collected and identified by morphology. These were sorted to 150 OTUs belonging to 6 orders of insects (Table 1). Among these specimens, 1995 were classified as phytophagous (1694 piercing-sucking and 301 chewing insects), 1839 as their natural enemies including 1358 as potential natural enemies of aphids.

No interaction was found between the inoculant treatments and the potassium at Varennes (S2 Table). There was a significant difference in the abundance of piercing-sucking insects by inoculation treatments irrespective of potassium application ($F_{2.28} = 4.12$, P = 0.026, S2 Table). The average abundance of piercing-sucking insects was higher with the inoculant [MRB] (26.1) relative to the control [C] (18.9), while the number of insects with the inoculant [MR] (22.8) was not different from that of other treatments (S3 Table). There were no differences among inoculant treatments for the abundance of other insect groups (chewing insects, aphids without *A. glycines*, aphids' natural enemies, *A. glycines* and *Empoasca* spp.) (S2 Table).

Similarly, there were no differences among inoculants treatments for the insect groups at Saint-Simon, except *A. glycines* and *Empoasca* spp. (S2 Table). The significant interaction between inoculants and potassium was observed with the abundance of *A. glycines* ($F_{2.21} = 6.69$, P = 0.006, S2 Table, Fig 1). In potassium-fertilized [K+] plots (blue boxplots), there was a

**Table 2. Arbuscular mycorrhizal fungi root colonization and yield of soybean at different levels of inoculation irrespective of potassium fertilizer application at Varennes and Saint-Simon.**

| Site | Plant parameters | Inoculant treatments | | | P |
|------|------------------|------|------|------|---|
| | | C | MR | MRB | |
| Varennes | AMF root colonization (%) | 43.8±2.82 | 40.6±3.34 | 39.3±4.04 | *ns* |
| | Yield (kg/ha) | 3060±35.86 | 3038±39.75 | 3016±35.66 | *ns* |
| Saint-Simon | AMF root colonization (%) | 60.2±4.63 | 61.3±4.09 | 64.2±4.02 | *ns* |
| | Yield (kg/ha) | 3075±84.08 | 3151±61.13 | 3162±71.32 | *ns* |

C: Control; MR: Mycorrhizae+Rhizobium; MRB: Mycorrhizae+Rhizobium+Bacillus. Values represent means ± SE of 8 replicates (n = 48 plots) in each site.

*ns*: Not significant.

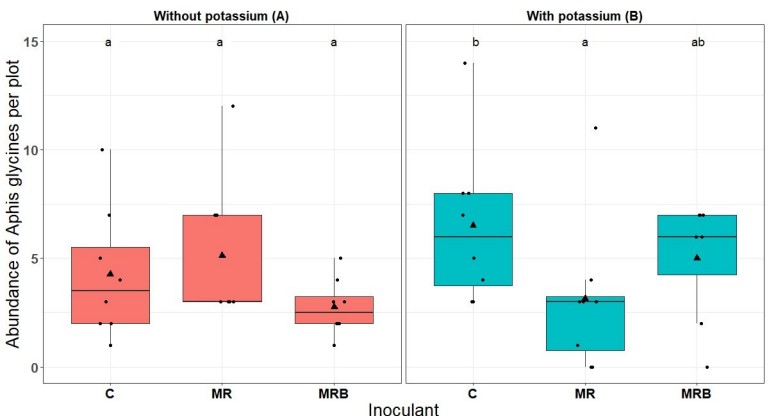

**Fig 1. Boxplots of *Aphis glycines* abundance per plant in each inoculant treatment (Control: C; Mycorrhizae +Rhizobium: MR; Mycorrhizae+Rhizobium+Bacillus: MRB) at Saint-Simon.** (A) with the application of potassium and (B) without the application of potassium. Letters above indicate significant differences among treatments based on Tukey's honest significant difference (HSD) test after Linear mixed effect model (LMM) follow by ANOVA (*P*<0.05). Triangle dots inside the boxplots represent the means.

significant difference between the control [C] plots, which had higher numbers of aphids relative to the inoculated [MR] plots, while aphid numbers in inoculated [MRB] plots were not different than the other treatments (Fig 1B). However, in the unfertilized [K-] plots (pink boxplots), the inoculations did not have any significant effect on the abundance of this aphid species (Fig 1A): all three inoculations yielded similar results. Independent of inoculant, *Empoasca* spp. exhibited a difference in the presence of potassium ($F_{1,35}$ = 5.53, P = 0.024, S2 Table). The plots without potassium [K-] had higher number of leafhoppers than the potassium-fertilized [K+] plots (S3 Table).

We examined the effect of the inoculants on the diversity of the second trophic level (i.e. the 36 phytophagous insect OTUs). As with the previous analyses on the abundance of insect groups, the various inoculant and fertilizer treatments showed no effect on phytophagous insect diversity, as measured with the Shannon index, neither at Varennes ($F_{2,35}$ = 0.66, P = 0.50) nor at Saint-Simon ($F_{2,21}$ = 0.57, P = 0.57) (S2 Table).

### 3.3. Correlation between AMF colonization and abundance/richness of insects

Even though it is not possible to discriminate the root colonization realized with the inoculum from that caused by indigenous AMF, the variation in the degree of mycorrhizal root colonization irrespective of inoculation treatments allowed us to test the effect of mycorrhization on insect abundance and diversity, independent of inoculant treatment. The abundance of piercing-sucking insects, chewing insects, as well as their alpha diversity, were all negatively correlated with the level of mycorrhizal colonization (as measured per plot) at Varennes, whereas no correlation was observed at Saint-Simon (Figs 2–4, S4 Table). No correlation was observed at either site between the level of mycorrhizal colonization and the abundance of natural enemies of aphids, *A. glycines*, aphids excluding *A. glycines*, or *Empoasca* spp. (S4 Table).

## 4. Discussion

The inoculation with *R. irregularis*, *B. japonicum* and *B. pumilus*, irrespective of potassium fertilizer in soybean fields showed differences in the abundance of piercing-sucking insects at

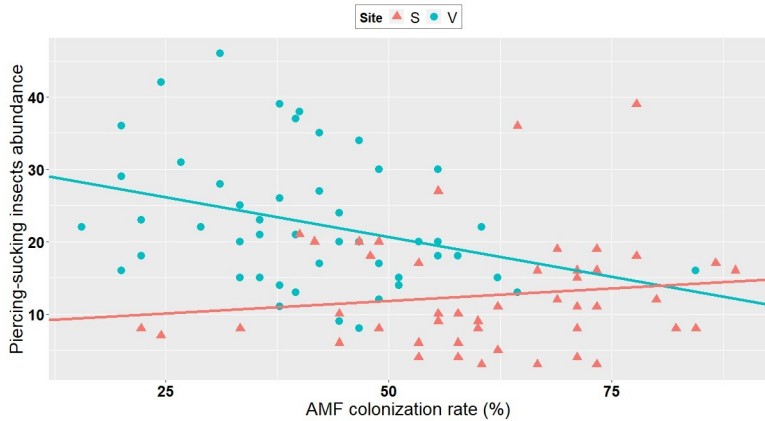

**Fig 2. Correlation between AMF colonization rate and piercing-sucking insect abundance at Varennes and Saint-Simon.** Negative correlation at Varennes (V; blue dots) (Kendall *tau*: -0.24; P = 0.016). No correlation at Saint-Simon (S; pink dots) (Kendall *tau*: 0.08; P = 0.43).

Varennes (S2 and S3 Tables), whereas no effect was observed on the abundance of chewing insects, aphids, aphid enemies, or on the species richness of phytophagous insects. At Saint-Simon, the inoculations had no discernible effect on any of these same insect groups (S2 Table), except for *Empoasca* spp. which exhibited difference in abundance in potassium-fertilized plots (S2 and S3 Tables). But with *A. glycines*, the inoculants interacted with potassium (Fig 1; S2 Table). Numerous studies have paid attention to the effects of belowground symbiotic microbes on aboveground plant–arthropod interactions [55,56]. For example, arbuscular mycorrhizal fungi positively affected the abundance of piercing-sucking insects and specialist chewers but decreased that of generalist chewers [30]. In the same line, some studies showed that the community composition of herbivores was significantly different between plants associated or unassociated with Rhizobia [33,57]. For example, Rhizobia positively affected the chewing insects but not the piercing-sucking insects [33]. Studies conducted on free-living PGPR reported negative effects of PGPR on pests in different crops [32]. However, most of these studies were conducted in controlled conditions with one or few strains. In field conditions, cultivated legumes encounter a diverse local rhizobacteria (Rhizobia and free-living

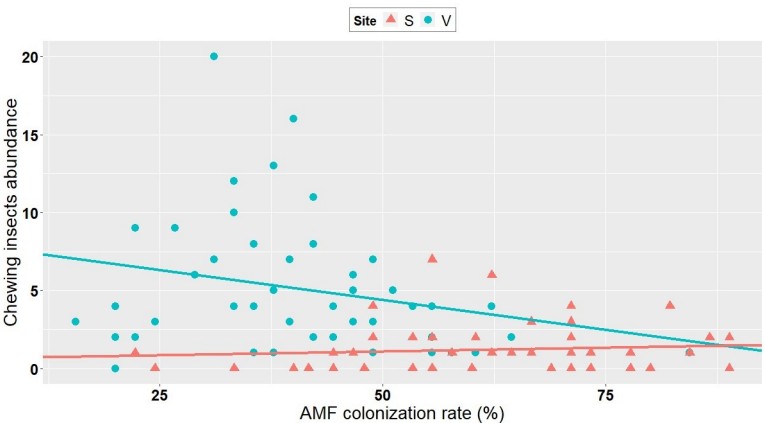

**Fig 3. Correlation between AMF colonization rate and chewing insect abundance at Varennes and Saint-Simon.** Negative correlation at Varennes (V; blue dots) (Kendall *tau*: -0.20; P = 0.052). No correlation at Saint-Simon (S; pink dots) (Kendall *tau*: 0.075; P = 0.50).

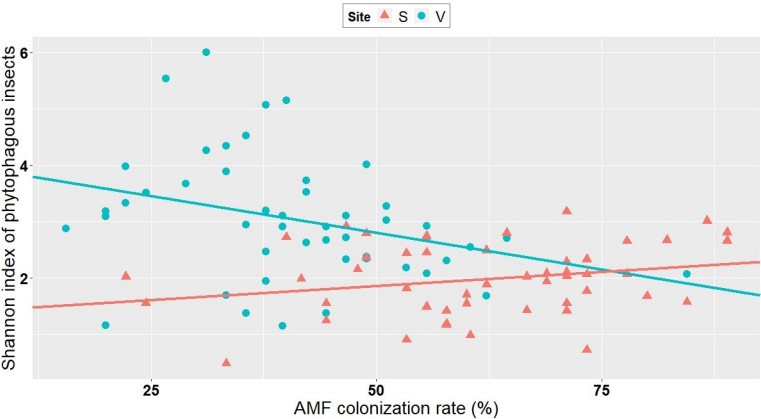

**Fig 4. Correlation between AMF colonization rate and Shannon index of phytophagous insects (piercing-sucking and chewing) at Varennes and Saint-Simon.** Negative correlation at Varennes (V; blue dots) (Kendall tau: -0.26; P = 0.008). No correlation at Saint-Simon (S; pink dots) (Kendall *tau*: 0.15; P = 0.11).

PGPR) and AMF community in the soil, including microorganisms from inoculants applied [30,35,57]. Further, both introduced inoculants and indigenous microbial community can influence host plant [36,58], and this can temper the dramatic effects otherwise seen in non-field conditions. In our study, the presence and diversity of indigenous AMF explains root colonization observed in the non-inoculated control plots (Table 2). However, our root colonization observations were purely quantitative, and thus do not preclude a possible compositional change in the root colonizing AMF community.

We found that the differences among inoculation treatments are not only affected by the sites of action, but also by the fertilization. On potassium-enriched plots in Saint-Simon, there was a significant difference between the control [C], which had the highest number of aphids, as compared to the double inoculant [MR], while no effects were observed in plots without potassium fertilizer (Fig 1B). The decrease we observed in soybean aphid abundance only with potassium fertilization suggests that the interaction between rhizobium and AMF can be influenced by nutrient condition. Previous studies showed that increased potassium levels directly and negatively influenced the soybean aphid [59,60]. It has been suggested that potassium deficiency in plant may induce an increase in levels of asparagine and other low-molecular-weight nitrogen-containing compounds, being thereby beneficial to aphids that have a nitrogen-limited diet [25]. However, in the current study, apart *Empoasca* spp., there was no difference between potassium treatments for all the insect groups, showing the previously reported scenario is not always the case in actual field conditions.

Importantly, our study found that independent of inoculation, AMF colonization seems to play an important role in microorganism-plant-insect interactions. In line with our second hypothesis, i.e, AMF colonization of plants can affect insect abundance and richness in field conditions, there was a negative correlation between the degree of AMF colonization and the abundance of (1) piercing-sucking insects, (2) chewing insects and (3) the species richness of phytophagous insects (Figs 2–4). Contrary to these findings, some reports have shown that increased AMF colonization positively influences insect performance [30,53,61,62]. Functional groups such as piercing-sucking insects (specialists and generalists) and specialist chewing insects were positively associated with high AMF-colonized plants [30]. These studies suggested that in well-established mycorrhizal plants, there was an increase of carbon/nutrient balance, which in turn lead to increased levels of carbon-based feeding deterrents, such as iridoid glycosides, that were less deleterious to piercing-sucking and specialist insects [30] than

to generalists. Specialists show a high degree of adaptation to their host's defenses and they usually perform better on mycorrhizal plants, probably because of the improved nutritional quality of the host [26].

On the other hand and in line with our investigation, some reports documented negative AMF-induced effects on insects [30]. Gange et al. (2002) showed that generalist chewing insects were negatively affected by the presence of well-established AMF communities [63]. Generalist chewing insects are relatively sensitive to plant defenses: they feed on leaf tissue, causing massive damage that activates a strong chemical defense [62]. Therefore, it is possible that the negative correlation between chewing insects and AMF plant colonization observed in our study is due to the generalist feeding habits of the chewing insects sampled. The fact that significant correlations were found only at Varennes and not at Saint-Simon is likely because the former site had an overall high number of these insect groups (Table 1).

Observed correlations in our study cannot answer which factors are actual causes or mere correlations. The outcome of an AMF-plant-insect interaction can also work in the reverse direction: herbivory can reduce AMF root colonization [29,62]. In a study where aphids were introduced to the plant before its being colonized by AMF, authors observed a reduction in subsequent AMF colonization [64]. They suggested that the antagonistic effect of aphids could operate either via reduced carbon allocation to AMF, because aphids drain carbon from the plants, or by defence-related signalling induced by the aphids that is antagonistic to AMF.

## 5. Conclusions

Our study showed that, under some field conditions, rhizosphere microbe inoculation can elicit an effect depending on the feeding mode of insects on soybean. However, understanding the more proximate causes of that inoculation effect will require further research on the relative diversity and composition of the microbial rhizosphere flora. Our key finding was that the mycorrhizal status of the plant plays a role in AMF-plant-insect interaction. We found that the abundance and richness of phytophagous insects (piercing-sucking and chewing) and their alpha diversity were negatively correlated with the AMF colonization rate. We suggest that mycorrhizal colonization plays a key role in insect-plant microbe interactions and its effects on insects depend on the degree of feeding specificity. Still there are contradictory reports regarding AMF effects on insect communities. Considering the suppressive effect of potassium fertilization on aphids in the AMF-inoculated plots and the other site- specific effects observed in our study suggest that, the abiotic environment also plays a crucial role in these tripartite interactions. Along with a better understanding of the many microbial actors, we believe future study on abiotic conditions will bring better understanding of these interactions and harnessing of microorganism for agriculture.

## Supporting information

**S1 Table. Arbuscular mycorrhizal fungi root colonization and yield of soybean at Varennes and Saint-Simon based on inoculation treatments (Control (C), Mycorrhizae+Rhizobium (MR), Mycorrhizae+Rhizobium+Bacillus (MRB)), potassium treatments (K-: Without potassium; K+: With potassium), tested individually and in interaction (*F-value, df, P-value*).** $P < 0.05$; n = 48.
(DOCX)

**S2 Table. Abundance of insect functional groups sampled on soybean at Varennes and Saint-Simon based on inoculation treatments (Control (C), Mycorrhizae+Rhizobium (MR), Mycorrhizae+Rhizobium+Bacillus (MRB), potassium treatments (K-: Without**

potassium; K+: With potassium), tested individually and in interaction (**F-value, df, P-value**). Linear mixed effect model (LMM) follows by ANOVA. *: $P < 0.05$; **: $P < 0.001$; $n = 48$.
(DOCX)

**S3 Table. Abundance of piercing-sucking insects and *Empoasca* spp. sampled on soybean at Varennes and Saint-Simon respectively based on inoculation treatments (Control (C), Mycorrhizae+Rhizobium (MR), Mycorrhizae+Rhizobium+Bacillus (MRB)) irrespective of potassium (K-: Without potassium; K+: With potassium) and vice versa.** Values represent mean ± SE of 8 replicates (n = 48) in each site. Letters follow by mean ± SE indicate significant differences among treatments based on the Tukey's honest significant difference (HSD) test after Linear mixed effect model (LMM) follow by ANOVA. *: $P < 0.05$.
(DOCX)

**S4 Table. Correlation between the rate of arbuscular mycorrhizal fungi colonization in the roots of soybean and the abundance and richness of insects at Varennes and Saint-Simon.** Values represent correlation coefficients. **: $P < 0.05$; *: $P < 0.1$.
(DOCX)

**S1 Data.**
(XLSX)

## Acknowledgments

We would like to thank Premier Tech Biotechnologies (Rivière-du-Loup, Quebec, Canada) for the support in field trials and experimental design, especially Dominique Léquéré. We also thank Renaud Hadd and Lucie Kablan from Sollio Cooperative Group for their assistance. We are gratefully to Thomas Théry (Université de Montréal) for his support in the identification of some insect species. We also thank Stéphane Daigle (Institut de recherche en biologie végétale) for help with statistical analyses. We gratefully acknowledge the helpful comments from anonymous referees that improved the quality of this manuscript.

## Author Contributions

**Conceptualization:** Élisée Emmanuel Dabré, Mohamed Hijri, Colin Favret.

**Data curation:** Élisée Emmanuel Dabré.

**Formal analysis:** Élisée Emmanuel Dabré.

**Funding acquisition:** Mohamed Hijri, Colin Favret.

**Methodology:** Élisée Emmanuel Dabré, Soon-Jae Lee.

**Writing – original draft:** Élisée Emmanuel Dabré.

**Writing – review & editing:** Soon-Jae Lee, Mohamed Hijri, Colin Favret.

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
