## [Decision Letter · Decision Letter 0]

26 Feb 2021

PONE-D-20-33487

The effects of rhizosphere microbiome inoculation and mycorrhizal infection on phytophagous insects in soybean fields

PLOS ONE

Dear Dr. Dabré,

Thank you for submitting your manuscript to PLOS ONE. After careful consideration, we feel that it has merit but does not fully meet PLOS ONE’s publication criteria as it currently stands. Therefore, we invite you to submit a revised version of the manuscript that addresses the points raised during the review process.

The manuscript has been reviewed by two experts. Please revise the manuscript following all the criticisms and prepare a detailed rebuttal letter answering point by point to the reviwers' comments. 

We look forward to receiving your revised manuscript.

Kind regards,

Raffaella Balestrini

Academic Editor

PLOS ONE

"EED received a scholarship from the Islamic Development Bank."

"MH: Natural Sciences and Engineering Research Council of Canada (NSERC) Discovery grant (RGPIN-2018-04178)

CF: Natural Sciences and Engineering Research Council of Canada (NSERC) Discovery grant (RGPIN 2017-06287)

Url: https://www.nserc-crsng.gc.ca/

"

Additionally, because some of your funding information pertains to commercial funding, we ask you to provide an updated Competing Interests statement, declaring all sources of commercial funding.

In your Competing Interests statement, please confirm that your commercial funding does not alter your adherence to PLOS ONE Editorial policies and criteria by including the following statement: "This does not alter our adherence to PLOS ONE policies on sharing data and materials.” as detailed online in our guide for authors  http://journals.plos.org/plosone/s/competing-interests.  If this statement is not true and your adherence to PLOS policies on sharing data and materials is altered, please explain how.

Please include the updated Competing Interests Statement and Funding Statement in your cover letter. We will change the online submission form on your behalf.

Reviewers' comments:

Reviewer's Responses to Questions

**Comments to the Author**

1. Is the manuscript technically sound, and do the data support the conclusions?

Reviewer #1: No

Reviewer #2: Yes

2. Has the statistical analysis been performed appropriately and rigorously? 

Reviewer #1: Yes

Reviewer #2: Yes

3. Have the authors made all data underlying the findings in their manuscript fully available?

Reviewer #1: Yes

Reviewer #2: Yes

4. Is the manuscript presented in an intelligible fashion and written in standard English?

Reviewer #1: Yes

Reviewer #2: Yes

5. Review Comments to the Author

Reviewer #1: The work titled “The effects of rhizosphere microbiome inoculation and mycorrhizal infection on phytophagous insects in soybean fields” investigates the employment of different inocula on insects in soybean fields.

Although the topic is interesting and the introduction is very nice, there is a big problem: the success of inoculation of bacteria has not been verified and the success of AMF is not so sure, seen that there are not differences in the mycorrhization rate among control, MR and MRB. The authors speak about a possible explanation for this result, but a doubt on AM inoculation failure remains. If the inoculation has not worked, here the authors are seeing the effects of the native population present in the soils on which there are not data. In this case all the hypothesis about inoculation are not valid. Other experiments or changes in the hypotheses to be tested would be necessary.

In my opinion, this paper is not suitable for publication in PlosOne. Here, there are some observations and suggestions for authors.

Major comments

Introduction

At line 24 there is a mistake. Glomeromycota is not a phylum but a subphylum on the base of Spatafora et al. 2016.

Material and methods

Why do the authors select these two sites with different sizes?

Could the inoculation dose explain the results of inoculation? Are there proofs about the success of this dosage in similar fields and with the same plant?

The authors speak about inoculation, but they find AMF in control….so it is not inoculation the right term. In the figure they speak about colonization that is more suitable. I suggest to modify the terminology in the manuscript.

I am not so sure that OTUs can be used when there is not molecular identification but only morphological. The current definition is this: the term "OTU" is also used in a different context and refers to clusters of (uncultivated or unknown) organisms, grouped by DNA sequence similarity of a specific taxonomic marker gene. It is difficult for me to understand the OTUs not in relation to DNA and level of insect identification.

Results

At lines 219-220 add the reference to data or table or supplementary or data not shown.

With the doubt about the success of inoculation all the results based on this are doubtful (3.1 and 3.2 paragraphs).

Discussion

At lines 318-319 the sentence is not right: the effects are not on treatments, but on variables.

In the discussion it is useful to put off to tables or figures.

At lines 350-352 authors speak about the effects of bacteria but they have never checked the inoculation success and persistence.

At line 424 they speak about high, but it is very difficult to say this, seen that there are not differences among the treatments.

Table

In Table 1 Piercing-sucking insects subtotal 20: which numbers do produce 20?

Figures

I think that Fig 2, 3 and 4 are not necessary, seen that there is information in Table 4.

Minor comments

At line 39 replace “dependent on” with “related to”.

At line 46 move “simultaneously” after can.

At line 53 add “s” at the verb.

At line 55 replace “One of” with “among” and add “there” before are.

At line 59 replace “reducing” with “limiting”.

At line 63 replace “quality” with “feature”.

At line 67 move the adverbs before affect and eliminate “be it”.

At line 70 add “also” after can.

At line 82 add “a” after of.

Lines from 114-116 replace “high and low” with maximum and minimum.

Check along the text not to separate subject and verb with comma.

At line 134 move “also” after the verb.

At line 141 add “insects” after sampled.

At line 149 replace “to” with “in” and move the adverb after was.

At lines 177 178 remove species and genus, and replace “these” with “this”.

At line 181 eliminate “measures of”.

At line 197 add “were studied using” and eliminate “, we used”.

At line 219 add “e” for here.

Move the comments in the results and put in the right section (lines 265-266; 282-283 moreover it is a sentence non clear).

At line 292 add “insects” after chewing.

At line 323 replace “we expected” with we can explain.

At line 324 replace “may be explained by” with “with”.

Chose as to write above-ground or aboveground.

At line 339-340 rephrase and add reference to table.

At line 349 replace “in high abundance” with abundantly.

At line 358 replace “most” with “majority the”.

At line 360 replace “relative” with compared.

At line 376 replace “that” with i.e.,

At line 430 there is some mistakes.

Reviewer #2: The manuscript entitled “The effects of rhizosphere microbiome inoculation and mycorrhizal infection on

phytophagous insects in soybean fields” by Dabré, Lee, Hijri and Favret deals with the effects on the abundance and the diversity of phytophagous bugs in two soybean fields, treated with the arbuscular mycorrhizal fungus, Rhizophagus irregularis, combined with a nitrogen-fixing bacterium, Bradyrhizobium japonicum, and a plant growth-promoting bacterium, Bacillus pumilus, in the presence and in the absence of potassium fertilizer. The topic is surely interesting, as the investigation was carried up in open fields on an economically important crop and it covers up to the third trophic level. The manuscript is clear and well written, statistical analyses and presented data are technically sound. Major drawbacks are the results of mycorrhizal colonization following inoculation with commercial inoculants, as they were similar with those obtained in control theses. Authors should revise the whole manuscript taking into account this issue, avoiding conclusive or deductive sentences on the effects due to inoculants (i. e. L267-268 change with something like “a significant higher number of insects was collected…” instead of “…the inoculants did affect abundance”). In this perspective, the correlation between insect abundance/diversity and mycorrhizal colonization rate (irrespective of the treatments) becomes therefore the main core of the manuscript. My suggestion is to shorten the title in “The effects of mycorrhizal infection on phytophagous insects in soybean fields”. From this point of view, discussion (especially the first part L318-331) needs to be reformulated, also because here and there it is a repetition of results.

In conclusion, taken together your data seem to indicate that higher abundance of pierce-sucking insects is observed with low mycorrhizal colonization rate, and also in the treatment MRB (not MR), at least in one site (table 3). An hypothesis could be that increase in insect abundance is due more to the plant growth-promoting bacterium, Bacillus pumilus, rather than the AMF. This might be discussed, with appropriate literature support.

The manuscript needs major revisions to be accepted for publication.

Minor comments

L24: Glomeromycota in italics

L35: Delete “Meanwhile” and the comma between the subject “free-living PGPR” and the verb “can promote”

L55-74: make a unique paragraph.

L184: avoid “versus”, there is no an opposition, only a distinction in different categories.

L216: add “measured in different plots” after “colonization rate. It should improve clarity of the analysis.

L219: here not her.

L219-224: Maybe the passive form for these sentences is more appropriate.

L219-220 and L224-225: these sentences are repeating the same concept.

L248: the data were pooled irrespective of K+/K- treatments, isn’t it? Please specify here and in caption to Table 3.

L290: add “irrespective of treatments” after “rate of mycorrhization”. It should improve clarity of the analysis.

L358-359: please rephrase, not very clear. Which “other effects”?

L370-371: use “similar” instead of “the same”, Empoasca spp. are known as mainy mesophyll feeder though not exclusive, aphids mainly phloem feeders.

6. PLOS authors have the option to publish the peer review history of their article (what does this mean?). If published, this will include your full peer review and any attached files.

Reviewer #1: No

Reviewer #2: No

---

## [Author Response · Author response to Decision Letter 0]

24 Apr 2021

Response to Reviewers, PLOS ONE 

Thank you for your time handling our manuscript. We here address the concerns raised by the reviews of the first version. Reference to specific line numbers here refer to the clean version of the revised manuscript (not the one with track changes). The most important limitation of the manuscript concerned the probable insufficiency of microbial inoculation since there was as much mycorrhizal infection in the control as in the experimental plots. To address this concern, we emphasized the uncertain nature of the proximate cause of the results of the inoculation experiment. That is, we explicitly acknowledge that the inoculation did not increase the level of mycorrhization and that we did not measure the microbial diversity (L215-219) and, therefore, that we can only surmise that these unevaluated factors played a role in (L305-307), for example, increasing the abundance of piercing-sucking insects in the triple-inoculant plots. As a consequence, we recommend that future research evaluate the full biological diversity of the rhizosphere (L20-21 and L373-376). We maintained the results section largely intact but, to refocus it more towards the evaluation of overall mycorrhization and away from the treatment experiment, we transferred former Tables 3 and 4 to the supplementary materiel. Finally, we revised the title of the manuscript to better reflect the content. 

We implemented the journal requirements: 1) style requirements; 2) methods section with additional information concern the fields site access (L135-137); 3) funding information that we removed from the Acknowledgments section and added to cover letter.

Here are the responses to the questions of reviewers.

Reviewer #1: “there is a big problem: the success of inoculation of bacteria has not been verified and the success of AMF is not so sure, seen that there are not differences in the mycorrhization rate among control, MR and MRB. The authors speak about a possible explanation for this result, but a doubt on AM inoculation failure remains. If the inoculation has not worked, here the authors are seeing the effects of the native population present in the soils on which there are not data. In this case all the hypothesis about inoculation are not valid. Other experiments or changes in the hypotheses to be tested would be necessary.”

- Thank you very much for your helpful criticism. We hope that the changes described in our paragraphs above address the reviewer’s concerns. We agree that the hypotheses regarding the inoculation treatments are not fully supported given that 1) the treatments did not visibly affect the rate of mycorrhization and 2) we did not measure the relative diversity of the microbial rhizosphere. We also agree that a better comprehension of the microbial rhizosphere in general and, especially in the context of the hypotheses we proposed, how it is affected by soil inocula, would help evaluate their effects on the multitrophic system we studied. However, even microbiologists admit to a lack of understanding of the microbial diversity of the rhizosphere and of the difficulties of studying it (Hepper et al. 1988; Dean et al. 2009; Engelmoer et al. 2014) and, unfortunately, we are not in a position to retrospectively evaluate the very complex microbial flora of our study system. Nonetheless, we think our limited results are worthy of being made available to interested readers because we did in fact detect an effect of inoculation, even if we cannot explain its proximate cause. We hope our experience will help inform future research.

Major comments

Introduction

At line 24 there is a mistake. Glomeromycota is not a phylum but a subphylum on the base of Spatafora et al. 2016.

- L28: indeed, Schüßler et al. (2001) introduced the phylum Glomeromycota to accommodate arbuscular mycorrhizal fungi. But Spatafora et al. 2016 introduced subphylum Glomeromycotina Spatafora & Stajich (demoted the phylum to subphylum) and accepted under phylum Mucoromycota. However, certain authors such as Tedersoo et al. 2018, Wijayawardene et al. 2018, maintained the phylum Glomeromycota. 

Material and methods

Why do the authors select these two sites with different sizes?

- L97-98: the experiment was done in collaboration with a company producing and promoting biofertilizers that also provided the living materiel. The sites and experimental design had been already established by the company (L135-137). 

Could the inoculation dose explain the results of inoculation? Are there proofs about the success of this dosage in similar fields and with the same plant? 

- Thank for your questions regarding inoculation. Indeed, the effects of plants inoculation with commercial strains in natural environment remains a challenge. Because in natural conditions, many of these organisms exist indigenously and the interaction with the plant depend on several factors: the quantity and viability of the strains in the soil, the agricultural practices that are unfavorable to the beneficial organisms, the soil, and climatic conditions.

- In L304-312 we explained this situation with the case of AMF because in the control plots, we had colonization independent of inoculation. This confirms that the soil microbiome can play an important role in the microorganism-plant relationships. You were right about the worries raised concern the attribution of the effects due to inoculation. But not having evaluated the mycorrhizal and bacterial component of the rhizosphere, we can make assumptions to explain the results. What we should focus more is the core of this study which remains the correlation between mycorrhizal infection and insect abundance and richness. 

- We are not aware of any published studies that have tested the relative success of various dosages in similar soybean fields.

The authors speak about inoculation, but they find AMF in control….so it is not inoculation the right term. In the figure they speak about colonization that is more suitable. I suggest modifying the terminology in the manuscript. 

- We reserved the term "inoculation" to refer to the commercial strains put into the soil at the time of seeding. Thank for your suggestion.

I am not so sure that OTUs can be used when there is not molecular identification but only morphological. The current definition is this: the term "OTU" is also used in a different context and refers to clusters of (uncultivated or unknown) organisms, grouped by DNA sequence similarity of a specific taxonomic marker gene. It is difficult for me to understand the OTUs not in relation to DNA and level of insect identification. 

- L168-171: you are right about the term “OTU" in a microbial context, but in entomology, operational taxonomic units (OTUs) can be used to indicate categories of specimens that share morphological characters: these are sometimes call morphospecies. We cited Favret et al 2019 as reference. 

Results

At lines 219-220 add the reference to data or table or supplementary or data not shown. With the doubt about the success of inoculation all the results based on this are doubtful (3.1 and 3.2 paragraphs).

- L213 and L225 respectively 3.1 and 3.2 paragraphs: as explain above, it is difficult to assert that inoculation did not work, even if it was not measurable, due to the certain presence of indigenous mycorrhizal spores. Of course, this is the difficulty of working in a natural setting instead of the controlled laboratory, but even if we cannot be definitive, we are still left to explain why the plots with triple-inoculant-treatments saw an increase in the number of piercing-sucking insects. 

- L304-312: we explain these results by a possible influence of each or both inoculants (introduced and native). For example, a study by Dean et al. 2009 (https://doi.org/10.1007/s11104-009-9924-1) compared 2 sources of Bradyrhizobium japonicum (commercial and native) on soybean aphid and showed a decrease in the aphid abundance with the native strain. Without a genetic analysis to get the identity of the strains, there could think that this effect was due to the introduced inoculants. 

- Because of the presence of native strains in our study and not having measured the microbial community in the soil, we think that it is hard to reject these results. 

Discussion

At lines 318-319 the sentence is not right: the effects are not on treatments, but on variables.

- L289-292: sentence changed. The term "treatments" referred to inoculant treatment with the 3 groups (control, MR and MRB) (S1 Table). 

- 

In the discussion it is useful to put off to tables or figures.

- L288-358: done.

At lines 350-352 authors speak about the effects of bacteria but they have never checked the inoculation success and persistence.

- L310-312: sentence changed.

At line 424 they speak about high, but it is very difficult to say this, seen that there are not differences among the treatments.

- L358: sentence removed; here, it is to specify that despite a measurable effect of inoculation, there was yet an important level of mycorrhizal colonization in roots (L308-310). Term changed.

Table

In Table 1 Piercing-sucking insects subtotal 20: which numbers do produce 20?

- L184 (Table 1): 20 is the sum of OTUs =13+1+6 respectively Aphids subtotal+Empoasca spp.+ Other piercing-sucking insects.

Figures

I think that Fig 2, 3 and 4 are not necessary, seen that there is information in Table 4.

- We kept the figures to show the direction of the correlation and put the Table 4 as supplementary material (S4 Table).

Minor comments (the responses are in bold)

At line 39 replace “dependent on” with “related to”- L40: word changed.

At line 46 move “simultaneously” after can- L47: word removed.

At line 53 add “s” at the verb- L52: letter added.

At line 55 replace “One of” with “among” and add “there” before are- L54: sentence changed after revision.

At line 59 replace “reducing” with “limiting”- L58: word replaced.

At line 63 replace “quality” with “feature”- L61: word replaced.

At line 67 move the adverbs before affect and eliminate “be it”- L64: sentence modified.

At line 70 add “also” after can- L68: word added.

At line 82 add “a” after of – L79: “a” added.

Lines from 114-116 replace “high and low” with maximum and minimum - L109-111: words replaced.

Check along the text not to separate subject and verb with comma: done

At line 134 move “also” after the verb- L129: word moved.

At line 141 add “insects” after sampled- L139: word added.

At line 149 replace “to” with “in” and move the adverb after was- L147: words replaced and moved.

At lines 177-178 remove species and genus and replace “these” with “this”- L174-175: words changed.

At line 181 eliminate “measures of”- L178: words removed.

At line 197 add “were studied using” and eliminate “, we used”- L196-197: sentence changed after revision.

At line 219 add “e” for here- L212: sentence removed after revision.

Move the comments in the results and put in the right section (lines 265-266; 282-283 moreover it is a sentence non clear)- L240-241: sentences changed.

At line 292 add “insects” after chewing- L270: word added.

At line 323 replace “we expected” with we can explain- L294: sentences changed.

At line 324 replace “may be explained by” with “with”- L295: sentences changed.

Chose as to write above-ground or aboveground- L294-295: suggestion done.

At line 339-340 rephrase and add reference to table- L308-309: sentence changed.

At line 349 replace “in high abundance” with abundantly- L310: sentence changed.

At line 358 replace “most” with “majority the”- L292-293: sentence changed.

At line 360 replace “relative” with compared- L323: sentence changed.

At line 376 replace “that” with i.e.,- L328: word replaced.

At line 430 there is some mistakes-L362: sentence corrected.

Reviewer #2: Major drawbacks are the results of mycorrhizal colonization following inoculation with commercial inoculants, as they were similar with those obtained in control theses. Authors should revise the whole manuscript taking into account this issue, avoiding conclusive or deductive sentences on the effects due to inoculants (i. e. L267-268 change with something like “a significant higher number of insects was collected…” instead of “…the inoculants did affect abundance”). In this perspective, the correlation between insect abundance/diversity and mycorrhizal colonization rate (irrespective of the treatments) becomes therefore the main core of the manuscript. My suggestion is to shorten the title in “The effects of mycorrhizal infection on phytophagous insects in soybean fields”. From this point of view, discussion (especially the first part L318-331) needs to be reformulated, also because here and there it is a repetition of results.

In conclusion, taken together your data seem to indicate that higher abundance of pierce-sucking insects is observed with low mycorrhizal colonization rate, and also in the treatment MRB (not MR), at least in one site (table 3). A hypothesis could be that increase in insect abundance is due more to the plant growth-promoting bacterium, Bacillus pumilus, rather than the AMF. This might be discussed, with appropriate literature support.

- Thank you for the kind words regarding the manuscript (not repeated here) and especially for the suggestions for its improvement. We have addressed the major criticism regarding the unimproved rate of mycorrhization in the experimental as opposed to the control plots in the replies to the editor and reviewer 1. Indeed, the introduction and especially the discussion were significantly modified.

Title of the manuscript

- L1-2: as suggested, we revised the title to emphasize the issue of mycorrhization correlated with the abundance and richness of phytophagous insects, which is the main core of the manuscript.

The whole manuscript

- Indeed, we noted that the question of how inoculations work is not explicit, and sometimes we went very quickly to certain conclusions, which was often confusing. In the revised manuscript, we provided more details about inoculations (introduction of commercial strains into the soil, which have been selected for their ability to enhance crop yields) and inoculants, which may be indigenous strains present in the soil in addition to the introduced strains. 

Results

L267-268

- As you suggested, we revised the manuscript by presenting our results with appropriate terms by avoiding deductive or conclusive sentences (i.e. L242-243).

Discussion

L318-331

- The first part of the discussion is reformulated considering the title and the results (L288-311). 

Taken together the data seem to indicate that higher abundance of pierce-sucking insects is observed with low mycorrhizal colonization rate, and also in the treatment MRB (not MR), at least in one site (table 3). A hypothesis could be that increase in insect abundance is due more to the plant growth-promoting bacterium, Bacillus pumilus, rather than the AMF. 

- L303-309: we agree with the reviewer that the bacterial components of the rhizosphere may have affected the insect populations. In light of this, we recommend further work be done to evaluate the taxonomic make-up of the microbial rhizosphere and its effects on the plants and higher trophic levels (L19-21). 

Minor comments (the responses are in bold)

L24: Glomeromycota in italics-L28: changed

L35: Delete “Meanwhile” and the comma between the subject “free-living PGPR” and the verb “can promote”- L39: meanwhile and the comma removed.

L55-74: make a unique paragraph-L54-72: done

L184: avoid “versus”, there is no an opposition, only a distinction in different categories-L181: “versus” delete and replace by “and”.

L216: add “measured in different plots” after “colonization rate. It should improve clarity of the analysis- L211: sentence modified as you suggested.

L219: here not her-L212: sentence removed after revision

L219-224: Maybe the passive form for these sentences is more appropriate- L212: sentence changed.

L219-220 and L224-225: these sentences are repeating the same concept-L215-216: sentence removed and corrected.

L248: the data were pooled irrespective of K+/K- treatments, isn’t it? Please specify here and in caption to Table 3-L233-234: yes, precision done and in caption to S1 Table.

L290: add “irrespective of treatments” after “rate of mycorrhization”. It should improve clarity of the analysis- L268: sentence modified as you suggested.

L358-359: please rephrase, not very clear. Which “other effects”?- L291-292: sentence changed.

L370-371: use “similar” instead of “the same”, Empoasca spp. are known as many mesophyll feeders though not exclusive, aphids mainly phloem feeders-L323: sentence removed after revision.

---

## [Decision Letter · Decision Letter 1]

17 May 2021

PONE-D-20-33487R1

The effects of mycorrhizal colonization on phytophagous insects and their natural enemies in soybean fields

PLOS ONE

Dear Dr. Dabré,

Thank you for submitting your manuscript to PLOS ONE. After careful consideration, we feel that it has merit but does not fully meet PLOS ONE’s publication criteria as it currently stands. Therefore, we invite you to submit a revised version of the manuscript that addresses the points raised during the review process.

Please still answer to the coments of reviewer # 2. 

We look forward to receiving your revised manuscript.

Kind regards,

Raffaella Balestrini

Academic Editor

PLOS ONE

Journal Requirements:

Reviewers' comments:

Reviewer's Responses to Questions

**Comments to the Author**

1. If the authors have adequately addressed your comments raised in a previous round of review and you feel that this manuscript is now acceptable for publication, you may indicate that here to bypass the “Comments to the Author” section, enter your conflict of interest statement in the “Confidential to Editor” section, and submit your "Accept" recommendation.

Reviewer #1: All comments have been addressed

Reviewer #2: All comments have been addressed

2. Is the manuscript technically sound, and do the data support the conclusions?

Reviewer #1: Yes

Reviewer #2: Yes

3. Has the statistical analysis been performed appropriately and rigorously? 

Reviewer #1: Yes

Reviewer #2: Yes

4. Have the authors made all data underlying the findings in their manuscript fully available?

Reviewer #1: Yes

Reviewer #2: Yes

5. Is the manuscript presented in an intelligible fashion and written in standard English?

Reviewer #1: (No Response)

Reviewer #2: Yes

6. Review Comments to the Author

Reviewer #1: The work ‘The effects of mycorrhizal colonization on phytophagous insects and their natural

enemies in soybean fields’ has been improved. I thank the authors for the added clarifications and done changes.

In my opinion, this paper is suitable for publication in Plos One.

Reviewer #2: The revision surely improves the manuscript quality that now is suitable for publication.

Minor issue:

L219 Change “microbial community is general” with “microbial community in general”.

7. PLOS authors have the option to publish the peer review history of their article (what does this mean?). If published, this will include your full peer review and any attached files.

Reviewer #1: No

Reviewer #2: No

---

## [Author Response · Author response to Decision Letter 1]

26 May 2021

Thank you for your time handling our manuscript. We hereby confirm that the reference list is update, complete and correct. Any reference cited in the text is in the bibliography.

Here is the response to the question of reviewer 2.

Minor issue: (the response is in bold)

L219- Change “microbial community is general” with “microbial community in general”- L219: word replace in the sentence

---

## [Decision Letter · Decision Letter 2]

29 Jul 2021

PONE-D-20-33487R2

The effects of mycorrhizal colonization on phytophagous insects and their natural enemies in soybean fields

PLOS ONE

Dear Dr. Dabré,

Thank you for submitting your manuscript to PLOS ONE. After careful consideration, we feel that it has merit but does not fully meet PLOS ONE’s publication criteria as it currently stands. Therefore, we invite you to submit a revised version of the manuscript that addresses the points raised during the review process.

Dear Dr. Dabré,

I was reassigned as editor for your paper yesterday; I apologize for the delay returning a review for your manuscript. The main issue that remains to be solved before this article can be published is the lack of clarity (and description) on the statistical analyses. You mention 6 treatments, but your analysis suggests a factorial design: C, MR and MRB crossed with K+ and K-. Did you test for an interaction between K and microbial treatments? Why not? Where is the block effect tested? Please report all stats (i.e. K treatment and block effects) for all tests; please also report df (missing in some tests). Why did you used a mixed model (i.e. what factors are random and what factors are fixed)?

It is unclear how some of the treatment comparisons were done. For example, in Lines 242-247, did you do comparisons of microbial treatments separately for the K+ and K- treatments? The df of 2 and 21 (L242) and 2 and 14 (L247) of the tests suggest that; please provide a more explicit description of the tests performed and the results obtained.

There are still several typos in the paper. I indicate several below, but it will be good if a another careful proof-read is made on the final version.

Minor changes:

L73 change “AM fungi” to “AMF” for consistency

L79: delete “in an agronomic system,” (sentence is too long, and at the end you mention soybean an under field conditions…)

L84: replace “is” by “are”

L123/129: fix typo in “block” (check for others I may have missed…) 

L198: the reference provided for the Shannon diversity index equation uses ln, not log2; please provide a reference using the formula you stated.

L201: change “responding variables” to “response variables”

L207: rewrite “In a case of significance difference was observed, “ to “When significant differences were observed,” ….

L211: please justify the use of Kendall correlation coefficients

L292: change “excepted” by “except for”

L341: change to “On the other hand and in line with our investigation,”

L346-348: change to “Therefore, it is possible that the negative correlation between

chewing insects and AMF plant colonization observed in our study is due to the generalist feeding habits of the chewing insects sampled”

Please revise all citations, several are incomplete, including: 12, 23, 24, 25, 38, 48, 58. Please follow journal format regarding the provision of doi (several are missing)

A rebuttal letter that responds to each point raised by the academic editor and reviewer(s). You should upload this letter as a separate file labeled 'Response to Reviewers'.A marked-up copy of your manuscript that highlights changes made to the original version. You should upload this as a separate file labeled 'Revised Manuscript with Track Changes'. **Please use red font color in all the changes in your manuscript, to facilitate the review (instead of track changes using word) **An unmarked version of your revised paper without tracked changes. You should upload this as a separate file labeled 'Manuscript'.

We look forward to receiving your revised manuscript.

Kind regards,

Alejandro Carlos Costamagna, Ph.D.

Academic Editor

PLOS ONE

Journal Requirements:

Additional Editor Comments (if provided):

Dear Dr. Dabré,

I was reassigned as editor for your paper yesterday; I apologize for the delay returning a review for your manuscript. The main issue that remains to be solved before this article can be published is the lack of clarity (and description) on the statistical analyses. You mention 6 treatments, but your analysis suggests a factorial design: C, MR and MRB crossed with K+ and K-. Did you test for an interaction between K and microbial treatments? Why not? Where is the block effect tested? Please report all stats (i.e. K treatment and block effects) for all tests; please also report df (missing in some tests). Why did you used a mixed model (i.e. what factors are random and what factors are fixed)?

It is unclear how some of the treatment comparisons were done. For example, in Lines 242-247, did you do comparisons of microbial treatments separately for the K+ and K- treatments? The df of 2 and 21 (L242) and 2 and 14 (L247) of the tests suggest that; please provide a more explicit description of the tests performed and the results obtained.

There are still several typos in the paper. I indicate several below, but it will be good if a another careful proof-read is made on the final version.

Minor changes:

L73 change “AM fungi” to “AMF” for consistency

L79: delete “in an agronomic system,” (sentence is too long, and at the end you mention soybean an under field conditions…)

L84: replace “is” by “are”

L123/129: fix typo in “block” (check for others I may have missed…)

L198: the reference provided for the Shannon diversity index equation uses ln, not log2; please provide a reference using the formula you stated.

L201: change “responding variables” to “response variables”

L207: rewrite “In a case of significance difference was observed, “ to “When significant differences were observed,” ….

L211: please justify the use of Kendall correlation coefficients

L292: change “excepted” by “except for”

L341: change to “On the other hand and in line with our investigation,”

L346-348: change to “Therefore, it is possible that the negative correlation between

chewing insects and AMF plant colonization observed in our study is due to the generalist feeding habits of the chewing insects sampled”

Please revise all citations, several are incomplete, including: 12, 23, 24, 25, 38, 48, 58. Please follow journal format regarding the provision of doi (several are missing)

Reviewers' comments:

Reviewer's Responses to Questions

**Comments to the Author**

1. If the authors have adequately addressed your comments raised in a previous round of review and you feel that this manuscript is now acceptable for publication, you may indicate that here to bypass the “Comments to the Author” section, enter your conflict of interest statement in the “Confidential to Editor” section, and submit your "Accept" recommendation.

Reviewer #1: All comments have been addressed

Reviewer #2: All comments have been addressed

2. Is the manuscript technically sound, and do the data support the conclusions?

Reviewer #1: Yes

Reviewer #2: Yes

3. Has the statistical analysis been performed appropriately and rigorously? 

Reviewer #1: Yes

Reviewer #2: Yes

4. Have the authors made all data underlying the findings in their manuscript fully available?

Reviewer #1: Yes

Reviewer #2: Yes

5. Is the manuscript presented in an intelligible fashion and written in standard English?

Reviewer #1: Yes

Reviewer #2: Yes

6. Review Comments to the Author

Reviewer #1: The work ‘The effects of mycorrhizal colonization on phytophagous insects and their natural

enemies in soybean fields’ has been improved too.

In my opinion, this paper is suitable for publication in Plos One.

Reviewer #2: All comments have been addressed.

7. PLOS authors have the option to publish the peer review history of their article (what does this mean?). If published, this will include your full peer review and any attached files.

Reviewer #1: No

Reviewer #2: No

---

## [Author Response · Author response to Decision Letter 2]

13 Aug 2021

Thank you for your time handling our manuscript. We here address the concerns raised by the reviews of the academic editor. Indeed, you were right about the statistics and thank you for your pertinent observations. We confirm that we performed an interaction between microbial treatments and potassium. Certainly, the way we presented the results could lead to believe that we had not performed it. Also, the lack of some details in our explanations may not allow for a better understanding. Thus, we revised the results tables with more clarity (L224-Table 2 and S Tables) and made a careful reading of this revised version of the manuscript. Reference to specific line numbers here refer to the clean version of the revised manuscript (not the one with track changes). 

Here are the responses to the comments and questions raised by the academic editor:

1-You mention 6 treatments, but your analysis suggests a factorial design: C, MR and MRB crossed with K+ and K-. 

L115-120-Indeed, it is a factorial design with 2 factors: an inoculant factor with 3 levels (C, MR and MRB) crossed with another factor, the potassium with 2 levels (K+ and K-); which gives 3x2 or 6 plots/bloc replicates 8 times.

2-Did you test for an interaction between K and microbial treatments? Why not?

Yes, we tested the interaction between microbial treatments and potassium. Maybe we were not precise, but the interaction was tested. Of all tests, only the interaction was observed with Aphis glycines at Saint-Simon (S Tables). That motivated us to make 2 tables (S1 effects of inoculants on insects and S2 effects of potassium). But in the revised manuscript we merged the 2 tables. Below figure is an example on the output of analysis of variance apply to the Linear mixed model (LMM) when testing the effects of inoculants and potassium on piercing-sucking insects at Varennes. It shows individual and the interaction effects between the two factors (inoculants and potassium).

aoVar_piq : LMM model 

aovVar_piq<-lmer(Tot_piq~Trait*K+(1|Bloc)+(1|Bloc:Trait)+(1|Bloc:K), data=Analys_Var)

Trait: inoculants treatments

K: potassium 

Trait:K= interaction

3-Why did you used a mixed model (i.e. what factors are random and what factors are fixed)? Where is the block effect tested? Please report all stats (i.e. K treatment and block effects) for all tests; please also report df (missing in some tests).

L200-201: we used mixed model because of fixed effects (inoculants treatments, potassium) and random effects (blocks). The block effects are included in the model formulation:

 Model<-lmer (variable~Trait*K+(1|Bloc) +(1|Bloc: Trait) +(1|Bloc: K), data=Data)

Trait=Inoculants K=Potassium

4-It is unclear how some of the treatment comparisons were done. For example, in Lines 242-247, did you do comparisons of microbial treatments separately for the K+ and K- treatments? The df of 2 and 21 (L242) and 2 and 14 (L247) of the tests suggest that; please provide a more explicit description of the tests performed and the results obtained.

In this paragraph, we would like to provide more details, but the explanations not sounded. Otherwise, there was a significant interaction between microbial treatments and potassium for the soybean aphid, Aphis glycines (F2.21=6.69, P=0.006, Fig 1; S1-S3 Tables). Later, we went into more details by showing the behavior of the microbial inoculants in the presence or absence of potassium, which effectively makes a separate analysis. In the revised manuscript, we corrected all the paragraph (L245-254).

Here are the responses to the minor issues raised by academic editor and reviewer (s).

5-Minor changes: (the response is in bold)

L73 change “AM fungi” to “AMF” for consistency-L73: word change

L79: delete “in an agronomic system,” (sentence is too long, and at the end you mention soybean an under field conditions…)-L79: sentence deleted

L82-L100: number of rhizobium bacterium strain corrected.

L84: replace “is” by “are”-L84: word replaced.

L123/129: fix typo in “block” (check for others I may have missed…)-L123/129: word corrected and in the whole manuscript.

L198: the reference provided for the Shannon diversity index equation uses ln, not log2; please provide a reference using the formula you stated-L199: you are right, it was an error and we corrected it.

L201: change “responding variables” to “response variables”-L203: sentence changed

L207: rewrite “In a case of significance difference was observed, “to “When significant differences were observed,”-L208-209: sentence rewritten.

L211: please justify the use of Kendall correlation coefficients-L211: We used Kendall correlation because our data did not meet all the assumptions of normal distribution to apply Pearson correlation that we made first.

L292: change “excepted” by “except for”- L299: word changed.

L341: change to “On the other hand and in line with our investigation,”-L348: sentence changed

L346-348: change to “Therefore, it is possible that the negative correlation between

chewing insects and AMF plant colonization observed in our study is due to the generalist feeding habits of the chewing insects sampled”-L353-354: sentence changed.

6-Please revise all citations, several are incomplete, including: 12, 23, 24, 25, 38, 48, 58. Please follow journal format regarding the provision of doi (several are missing) 

We hereby confirm that the reference list is update, complete and correct. Any reference cited in the manuscript is in the bibliography.

L426: citation 12- is a book-replaced by another citation also pertinent 

L463: citation 23- there is no doi associated with this citation on the journal site; but it has been completed by adding the journal name, the volume number, the year and the pages number (suggested citation)

L466: citation 24: there is no doi associated with this citation on the journal site; but it has been completed by adding the journal name, the volume number, the year and the pages number (suggested citation)

L469: citation 25-citation corrected 

L515: citation 38- became 39- corrected 

L549: citation 48-became 49- corrected

L579: citation 58-became 59-corrected

---

## [Editor Report · Decision Letter 3]

31 Aug 2021

PONE-D-20-33487R3

The effects of mycorrhizal colonization on phytophagous insects and their natural enemies in soybean fields

PLOS ONE

Dear Dr. Dabré,

Thank you for submitting your manuscript to PLOS ONE. After careful consideration, we feel that it has merit but does not fully meet PLOS ONE’s publication criteria as it currently stands. Therefore, we invite you to submit a revised version of the manuscript that addresses the points raised during the review process.

Thanks for the revised version of our paper and providing more details on your statistical analysis. There are few clarifications in the text of the manuscript needed before final publication.

Section 3.1. There are no Anova test results reported for the main effects, Potassium and inoculation treatments, and their interaction. Please report them in the text. In Table 2 you report tests comparing the inoculant treatments (C, MR, and MRB) separately within each K treatment. This is justified if you have a significant interaction between inoculant and K treatments, however that was not indicated. I think is ok to present the means+/- SE of the 6 treatments, as you have them currently in Table 2, but not the anova tests within factors unless you have significant interactions (in which case you also need to check differences for K within each inoculation treatment). Also, always report both degrees of freedom for your anova tests, numerator and denominator, separated by commas in your tables or as subscripts after the F in the text.

Supplementary tables: do not repeat values in tables S1 and S2. I suggest you delete the stats in table S2 (already reported in S1) and only leave the different letters indicating the Tukey test for pierce-sucking insects at Varennes and for the main effect test on Empoasca in Saint-Simon. I suggest you delete S3, as the same information is presented in Figure 2.

Minor changes:

L 194: replace “normal logarithm” by “natural logarithm”

L 238: F test number does not match exactly table S1, please check.

L 296/300/etc.; global change: please change “S1 and S2 Tables” to “Tables S1 and S2”

L 310: replace “fields conditions” by “field conditions”

L313: to avoid repetition with the previous sentence, I would replace “In field conditions,” by “Further,”

L357: replace insects’, by “insect”

L374: delete “the” before mycorrhizal

L375: delete “can” before depend

L385: after “in our study,” add “suggest that”

A rebuttal letter that responds to each point raised by the academic editor and reviewer(s). You should upload this letter as a separate file labeled 'Response to Reviewers'.A marked-up copy of your manuscript that highlights changes made to the original version. You should upload this as a separate file labeled 'Revised Manuscript with Track Changes'. **Please use red font color in all the changes in your manuscript, to facilitate the review (instead of track changes using word) **An unmarked version of your revised paper without tracked changes. You should upload this as a separate file labeled 'Manuscript'.

We look forward to receiving your revised manuscript.

Kind regards,

Alejandro Carlos Costamagna, Ph.D.

Academic Editor

PLOS ONE
---

## [Author Response · Author response to Decision Letter 3]

4 Sep 2021

Response to Reviewers, PLOS ONE 

We thank you for your important comments and suggestions. We here address the concerns raised by the reviews of the academic editor. Reference to specific line numbers here refer to the clean version of the revised manuscript (not the one with track changes). 

Here are the responses to the comments and questions raised by the academic editor:

1- Section 3.1. There are no Anova test results reported for the main effects, 

Potassium and inoculation treatments, and their interaction. Please report them in the text. In Table 2 you report tests comparing the inoculant treatments (C, MR, and MRB) separately within each K treatment. This is justified if you have a significant interaction between inoculant and K treatments, however that was not indicated. I think is ok to present the means+/- SE of the 6 treatments, as you have them currently in Table 2, but not the anova tests within factors unless you have significant interactions (in which case you also need to check differences for K within each inoculation treatment). Also, always report both degrees of freedom for your anova tests, numerator and denominator, separated by commas in your tables or as subscripts after the F in the text. 

Response: L216-229: Section 3.1. We added a new table named S1 Table (Supplementary materiel) with the Anova test results for the main effects, potassium, inoculation treatments, and their interaction. We also keep the Table 2 with means because we mentioned it as support in the discussion section (L315). For the degrees of freedom as numerator and denominator with F values, we reported them in all the text.

2. Supplementary tables: do not repeat values in tables S1 and S2. I suggest you delete the stats in table S2 (already reported in S1) and only leave the different letters indicating the Tukey test for pierce-sucking insects at Varennes and for the main effect test on Empoasca in Saint-Simon. I suggest you delete S3, as the same information is presented in Figure 2.

Response- With the insertion of the new S1 table, the former Tables S1 and S2 are know Tables S2 and S3 respectively in the revised version. In the new S3 Table, we deleted the stats and left only piercing-sucking insects at Varennes and Empoasca spp. at Saint-Simon. We also deleted the former S3 Table as you suggested.

3-Minor changes: (the response is in bold)

L 194: replace “normal logarithm” by “natural logarithm”-L194: word replaced

L 238: F test number does not match exactly table S1, please check-L238: F test number changed

L 296/300/etc.; global change: please change “S1 and S2 Tables” to “Tables S1 and S2”-L295-300: words changed

L 310: replace “fields conditions” by “field conditions”-L309: words replaced

L313: to avoid repetition with the previous sentence, I would replace “In field conditions,” by “Further,”-L312: sentence replaced by the suggested one

L357: replace insects’, by “insect”-L356: word replaced

L374: delete “the” before mycorrhizal-L373: word deleted

L375: delete “can” before depend-L374: word deleted

L385 (You mean L378? because the L385 is in the section of Acknowledgements): after “in our study,” add “suggest that”-L377: sentence added

---

## [Editor Report · Decision Letter 4]

9 Sep 2021

The effects of mycorrhizal colonization on phytophagous insects and their natural enemies in soybean fields

PONE-D-20-33487R4

Dear Dr. Dabré,

We’re pleased to inform you that your manuscript has been judged scientifically suitable for publication and will be formally accepted for publication once it meets all outstanding technical requirements.

Kind regards,

Alejandro Carlos Costamagna, Ph.D.

Academic Editor

PLOS ONE
---

## [Editor Report · Acceptance letter]

13 Sep 2021

PONE-D-20-33487R4 

The effects of mycorrhizal colonization on phytophagous insects and their natural enemies in soybean fields 

Dear Dr. Dabré:

I'm pleased to inform you that your manuscript has been deemed suitable for publication in PLOS ONE. Congratulations! Your manuscript is now with our production department. 

Kind regards, 

on behalf of

Dr. Alejandro Carlos Costamagna 

Academic Editor

PLOS ONE